# Tunable Thermo-Responsive Copolymers from DEGMA and OEGMA Synthesized by RAFT Polymerization and the Effect of the Concentration and Saline Phosphate Buffer on Its Phase Transition

**DOI:** 10.3390/polym11101657

**Published:** 2019-10-11

**Authors:** Alejandro Ramírez-Jiménez, Kathleen Abigail Montoya-Villegas, Angel Licea-Claverie, Mirian Angelene Gónzalez-Ayón

**Affiliations:** 1Cátedras CONACyT-Centro de Graduados e Investigación en Química, Tecnológico Nacional de México/Instituto Tecnológico de Tijuana, Blvd. Alberto Limón Padilla S/N, Ciudad Industrial Mesa de Otay, 22500 Tijuana, Baja California, Mexico; 2Centro de Graduados e Investigación en Química, Tecnológico Nacional de México/Instituto Tecnológico de Tijuana, Blvd. Alberto Limón Padilla S/N, Ciudad Industrial Mesa de Otay, 22500 Tijuana, Baja California, Mexico; kathleen.montoya@tectijuana.edu.mx (K.A.M.-V.); aliceac@tectijuana.mx (A.L.-C.);

**Keywords:** thermo-responsive polymers, RAFT, OEGMA, chain transfer agent, smart polymers, trithiocarbonate

## Abstract

Thermo-responsive polymers and copolymers derivatives of oligo(ethylene glycol) methyl ether methacrylate (*M_n_* = 300 g mol^−1^) (OEGMA) and di(ethylene glycol) methyl ether methacrylate (DEGMA) have been synthesized by reversible addition fragmentation chain transfer polymerization (RAFT) using 5-amino-4-methyl-4-(propylthiocarbonothioylthio)-5-oxopentanoic acid (APP) as chain transfer agent (CTA). The monomer conversion was evaluated by hydrogen nuclear magnetic resonance (^1^H-NMR); number average molecular weights (*M_n_*), weight average molecular weight (*M_w_*), and dispersity (*Đ*) were obtained by gel permeation chromatography (GPC); glass transition temperature (*T_g_*) was evaluated by modulated differential scanning calorimetry (DSC), cloud point temperature (*Tcp*) was measured and compared by turbidimetry and dynamic light scattering (DLS). The effect of polymer composition and concentration on the *Tcp*, either in water or in phosphate buffer saline (PBS), was studied. The values of *Tcp* using PBS were between 3 and 4 °C lower than using water. Results showed an ideal copolymerization behavior; therefore, the *Tcp* could be tuned by an adequate monomers feed ratio obtaining polymers which may be used in drug delivery and other applications.

## 1. Introduction

Thermo-responsive polymers which show an abrupt and reversible change in some properties within a small change of temperature, have been gaining much attention due to the wide variety of potential applications such as in drug delivery systems, tissue engineering, controlled cell or bacteria adhesion, protein separation, and sensors [1,2,3,4,5,6]. These polymers, generally in aqueous solution, show either a lower critical solution temperature (LCST), an upper critical solution temperature (UCST), or even both [7]. Several types of polymers are well known for this behavior [8,9], among these oligo(ethylene glycol) methyl ether methacrylates (OEGMAs), di(ethylene glycol) methyl ether methacrylate (DEGMA), as well as the analogue acrylate based polymers, have attracted great attention due to its resemblance to polyethylene glycol. Therefore, polymers of DEGMA and OEGMA with different molecular weighs have been widely investigated [10]. Perhaps the main advantage of these polymers over others is that, the temperature at which the phase transition occurs can be modified by tuning the hydrophobic-hydrophilic balance without changing the chemical nature of the components.

There are several reports where the phase transition temperature was studied as function of ethylene glycol units finding that the more ethylene glycol units the higher the transition temperature [2,11,12]. On the other hand, there are expectations suggesting that a well-defined structure of these kinds of polymers would lead to a better control of the phase transition temperature and a narrow range where this transition occurs. OEGMAs based polymers have been synthesized by anionic polymerization [13,14] and reversible deactivation free-radical polymerization like reversible addition-fragmentation chain-transfer polymerization (RAFT) or atom transfer radical polymerization (ATRP) [15,16,17,18,19,20]. Each of these options have advantages and disadvantages, however; owing to its scientific and practical relevance, RAFT polymerization has become one of the most applied techniques to obtain polymers with a narrow dispersity of molecular weights. This can be reached only by choosing the appropriate reaction conditions such as the temperature, concentration of monomers, and the monomer: chain transfer agent (CTA):initiator mole ratio.

RAFT is a versatile process because it can be carried out both in solution and in heterogeneous media as suspension or emulsion. Also, a wide type of monomers can be polymerized in dissolution using organic solvents or water depending of its hydrophilic character. In addition, RAFT polymerization has a high tolerance to different functional groups [21], therefore, different complex architectures such as blocks, stars, hyperbranched polymers, microgels, and supramolecular assemblies can be obtained. Many studies have been reported dealing with its application and its mechanism to the synthesis of novel materials. Excellent reviews and compilations have been published showing the versatility and fundamentals of this process [22,23,24].

Although there are several reports of the synthesis of DEGMA, OEGMA, and its copolymers using different methods and its thermo-responsive behavior is well known, a comparison of the results is not easy. There are many differences, for example, in molecular weight of the samples, its concentration in water, the method used for phase transition temperature determination, the use of different salts and its concentrations, the morphologies of the polymers obtained or even the role of different terminal groups. Due to the above; the aim of this work is to show the controlled synthesis of tunable thermo-responsive OEGMA and DEGMA based copolymers by RAFT polymerization, to evaluate its thermo-responsive behavior at different concentrations both in water and PBS, and to compare the obtained values using different methods of cloud point temperature (*Tcp*) determination like dynamic light scattering (DLS) and turbidimetry.

## 2. Materials and Methods

### 2.1. Materials

Poly(ethylene glycol) methyl ether methacrylate *M_n_* = 300 g mol^−1^ (OEGMA, Aldrich Chemical Co. Toluca, Edo. México, México) and di(ethylene glycol) methyl ether methacrylate (DEGMA, Aldrich Chemical Co.), were purified by passing through a column packed with inhibitor remover for hydroquinone and monomethyl ether hydroquinone (Aldrich Chemical Co.) and neutral alumina (Productos Químicos Monterrey, Monterrey, N.L. México) before use. Propane-1-thiol (Aldrich Chemical Co., ≥99%), carbon disulfide (Aldrich Chemical Co., ≥99.9%), sodium hydroxide pellets (Productos Químicos Monterrey 97.8%), acetone (J.T. Baker, Xalostoc, Edo. México, México 99.6%), ethyl acetate (Fermont, Monterrey, N.L. México 99.9%), methylene dichloride (Fermont ≥ 99.9%), diethyl ether (Fermont, ≥99.9), tetrahydrofuran (Aldrich Chemical Co. ≥99.9), potassium iodide (Fermont 99.8%), iodine (Aldrich Chemical Co. ≥ 99.8%), 4,4-azobis(4-cyanovaleric acid) (ACVA) (Aldrich Chemical Co., ≥98%), and anhydrous magnesium sulfate (Fermont, 99.8%) were used as received.

### 2.2. Equipment

FTIR-ATR spectra were recorded using a Perkin Elmer Spectrum 400, FTIR-NIR spectrometer (Perkin Elmer Cetus Instrument, Norwalk, CT) with eight scans between 650 and 4000 cm^−1^. Differential scanning calorimetry (DSC) was carried out using an Q2000 TA-instrument modulates DSC equipment (TA Instrument, New Castle, DE, USA), two heating cycles were recorded in modulation mode under a nitrogen flow of 50 mL min^−1^ at a heating rate of 5 °C min^−1^ with an amplitude of ±1 °C over a modulation period of 60 s. In both cycles the temperature was equilibrated at −80 °C during 5 min before the heating up to 180 °C for the first cycle and up to 120 °C for the second one, glass transition temperature (*T_g_*) was calculated by using the Universal Analysis software from TA Instrument. ^1^H and ^13^C-NMR spectra were recorded using a Brucker NanoBay equipment Ascend Magnet Spectrometer (Billerica MA, USA) (400.13 and 100.62 MHz, respectively) deuterated chloroform (CDCl_3_, 98%) or deuterated dimethyl sulfoxide (DMSO-d6) were used as solvents, the reported chemical shifts (ppm) are relative to (CH_3_)_4_Si. The number averaged molecular weight (*M_n_*) and dispersety index (*Đ*) were determined by gel permeation chromatography (GPC) using a Waters 515 HPLC chromatograph equipped (Malvern, Worcestershire, UK) with three columns in series (Shodex Asahipak, GF-1G 7B, GF-510 HQ, and GF-310 HQ) and two detectors: Refractive index detector (Wyatt Technology, Santa Barbara, CA, USA), operating at a wavelength of 633 nm, and a multiple light scattering detector (18 detector Dawn DSP, Wyatt Technology). The measurements were performed in methanol at 35 °C, polyethylene oxide standards were used for calibration. Electronic impact mass spectra (EIMS) were obtained using an Agilent 5975 mass (Santa Clara, CA, USA). Electrospray mass spectrometry (ESI) was performed using a Jeol AccuTOF JMS-T100LC mass spectrometer (Peabody, MA, USA) using ionization mode DART+. *Tcp* was measured using a Hach DR/890 portable colorimeter (Loveland, CO. USA), 15 mL of aqueous polymer dissolutions at different concentrations (mg mL^−1^) were prepared using deionized water or PBS; transmittance at 542 nm was measured as a function of temperature, transmittance 100% was fixed at the start temperature, *Tcp* was taken when transmittance reached 50%; for selected samples for selected samples the *Tcp* was measured by DLS using a Malvern Zata Sizer nano ZEN3690; Malvern Instruments, Miami, FL) equipped with a red laser (630 nm), the angle of measurement was 90°; the samples were allowed to equilibrate for 4 min for each temperature prior to measurement. The distribution by intensity using CONTIN analysis is reported. The hydrodynamic diameters (*D_h_*) were calculated using the Strokes-Einstein equation for spheres (Equation (1)), as usual.
*R_h_* = *k_B_T*/6π*η*_0_*D*(1)
where *k_B_* is the Boltzmann constant, *T* is the absolute temperature, *η*_0_ is the solvent viscosity, and *D* is the diffusion coefficient determined from DLS data. The effect of the temperature on the particle size was studied at concentration of 2 mg min^−1^ in the range between 20 and 50 °C; the *Tcp* was taken at the onset point where size was increased. The elemental analysis was obtained on a Thermo Scientific Flash-2000 (San Jose, CA, USA) at 950 °C, methionine was used as reference. UV-Vis spectra were recorded using an Agilent Cary 100 UV-Vis Spectrophotometer (Santa Clara, CA, USA) at wavelengths between 250 and 800 nm.

### 2.3. Synthesis of 5.amino-4-methyl-4-(propylthiocarbonothioylthio)-5-oxopentanoic Acid (APP) 

5-amino-4-methyl-4-(propylthiocarbonothioylthio)-5-oxopentanoic acid (APP) (Figure 1) was synthesized according to our previous report [25]. 1.027 g, 3.41 mmol of 4-cyano-4-(propylsulfanylthiocarbonyl)sulfanyl pentanoic acid (CPP) was dissolved in the minimum amount of methylene dichloride, then 100 μL of HCl = 0.1 N were added, the mix was left in an open container until slow solvent evaporation. The remaining solid was dissolving in acetone, filtered off to remove impurities, the sample was dried by slow solvent evaporation obtaining 0.587 g, 1.986 mmol, yield: 54% of a crystalline yellow solid, mp 163 °C (DSC), FTIR-ATR (cm^−1^): 3450 (m; *ν*(CO-OH)), 3318, 3252 (m; *ν*(NH_2_)), 2964, 2930 (m; *ν*(CH_3_, CH_2_)), 2873 (m; *ν*(CH_3_, CH_2_)), 1711 (vs; *ν*(CO-OH)), 1639 (vs; *ν*(CO-NH_2_)), 1068 (s; *ν*(C=S)), 804 (s; *ν*(C-S)); hydrogen nuclear magnetic resonance (^1^H-NMR) (400.13 MHz, DMSO-_d6_, *δ*): 12.23 (br s, 1H; COOH), 7.43 (s, 1H; C=O-NH-a), 7.24 (s, 1H; C=O-NH-b), 3.28 (t, J = 7.26 Hz, 2H; H-8), 2.17–2.29 (m, H-2, 4H; H-3); 1.59–1.73 (m, 5H; H-6, H-9); 0.94 (t, J = 7.36 Hz, 3H; H-10); ^13^C-NMR (100.62 MHz, DMSO-_d6_, *δ*): 221.8 (C-7), 174.0 (C-1), 172.5 (C-5), 61.7 (C-4), 38.6 (C-8), 32.9 (C-3), 29.8 (C-2), 24.3 (C-6), 22.7 (C-9), 21.5 (C-6), 13.6 (C-10); UV-Vis (acetone) λ max, nm, 214 (n → σ*); 327 (π → π); 450 (n → π*); EIMS (m/z (%)): 295 (1) [M^+^], 288 (6) [M-18]^+^, 219 (7), 172 (9), 144 (100); (ESI^+^, m/z): 296 (10) [M+H]^+^, 279 (100) [M-17]^+^; HRMS (ESI^+^, m/z): calculated for C_10_H_18_NO_3_S_3_, 296.04489; found for C_10_H_18_NO_3_S_3_, 296.04614; Analytically calculated for C_10_H_17_NO_3_S_3_·0.5(CH_3_)_2_CO: C 42.57, H 6.21, N 4.32, S 29.65; found: C 42.28, H 6.06, N 4.38, S 28.82.

### 2.4. Polymer Synthesis

Polymers of OEGMA or DEGMA and OEGMA-co-DEGMA were synthesized via RAFT (Scheme 1), 15 mmol of total monomers, 0.15 mmol of APP, and 0.038 mmol of initiator (ACVA) were placed into 20 mL glass ampoules using THF as solvent at 25% (*v*/*v*, %) of monomers total concentration, then argon was bubbled during 20 min and each ampoules was sealed, all reactions were placed during 4 h into an oil bath preheated at 70 °C, after of the reaction time, the ampoules were placed into an ice bath and opened, solvent was removed by a rotator evaporator, in order to calculate the monomer conversion, non-further purified samples were analyzed by FTIR-ATR and ^1^H-NMR, for the last one, around 20 mg were dissolved into deuterated solvent. 

Monomer conversion was calculated by ^1^H-NMR using the Equation (2).*Conv* (%) = 100% × *P*/(*P* + *M*)(2)
where *P* is the integration of the signals corresponding to the methylene group adjacent to ester group from the polymer which appeared between 4.02 and 4.21 ppm and *M* is the integration of the signal corresponding to the methylene group adjacent to ester group from residual monomers which appeared between 4.30 and 4.33 ppm. The polymers were purified by co-solvents precipitation; crude products were dissolved using methylene dichloride and precipitated into diethyl ether. The pure products were characterized by ^1^H-NMR, FTIR-ATR, DSC, and GPC, *Tcp* was determined by turbidimetry and DLS.

## 3. Results and Discussion

Trithiocarbonate type compounds have been used as CTAs in RAFT polymerization; these offer some advantages with respect to other CTAs, for example more stability toward hydrolysis [26]. CTAs containing 4-cyanovalec acid as R leaving group have been used widely, however; the nitrile may be auto-hydrolyzed to lead the corresponding amide. This was observed by Fuchs and Thurecht; they reported differences in the polymers obtained when a small fraction of CTA is hydrolyzed but, they did not separate these compounds [27]. In our previous report, the kinetic study of OEGMAs polymerization using both the isolated hydrolyzed and the not hydrolyzed CTAs was carried out, the second one was obtained as a crystalline solid and it was more stable [25], therefore this CTA was used in the current study.

In order to determine the control in the RAFT polymerization using APP, kinetic studies of DEGMA and OEGMA polymerization were carried out; in both cases an induction time period was observed which indicates that the pre-equilibrium in the RAFT process was not reached instantaneously (Figure 2a). Under the same conditions, DEGMA polymerization was faster than OEGMA polymerization, this may be due to the shorter side chains in DEGMA therefore; steric hindrance and diffusion affect the kinetics of reaction of this kind of polymers [28]. It is important to mention that in this study, the reaction was made in an organic medium because of when the process is carried out in aqueous solution the opposite behavior may be observed [29]. The pseudo-first order kinetic behavior with respect to the monomer was corroborated by a lineal relationship between ln[*M*_0_/*M*] versus time; the plots are showed in the Figure 2b. On the other hand, high molecular weights are reached in the beginning of the processes, this has been attributed to a hybrid behavior [30,31,32] where a conventional free radical polymerization (FRP) occurs in the early steps and once the RAFT pre-equilibrium is reached, the molecular weight shows an increase with respect to the conversion, linear regression for PDEGMA had an *R*^2^ value of 0.9138 however for POEGMA this value was too low (*R*^2^ = 0.7494). In the Figure 3a,b this behavior is observed; it is important to mention that the dispersity of molecular weights was lower for PDEGMA than for POEGMA.

Theoretical molecular weight was calculated using the Equation (3).
*M_n, Theo_* (g mol^−1^) = [M] × *Conv* × (*M*_Wm1_ + *M*_Wm2_)/[CTA] + *M*_WCTA_(3)
where [M] and [CTA] are the monomers and CTA concentrations, respectively, *Conv* is the fraction of monomers conversion, *M*_Wm1_ and *M*_Wm2_ are the products of the molecular weight from each monomer and its mol fraction in the reaction, and *M*_WCTA_ is the molecular weight of the CTA.

Molecular weights of PDEGMA obtained by RAFT polymerization were obtained by GPC, in the respective chromatograms unimodal distributions were obtained as shown in Figure 4. A decrease in the elution time can be seen by increasing the reaction time in the polymerization of DEGMA, this it is because of the polymer chains grow as the reaction times increases.

The GPC chromatograms for the copolymerization of DEGMA with different amounts of OEGMA are shown in Figure 5. The elution times were longer for the copolymers with less amount of OEGMA in their molar feed composition, with narrower distributions, the presence of a shoulder in the distribution curves of the copolymers is observed with greater clarity in those containing a higher amount of OEGMA in its composition, this can be attributed to the less control of the APP in the polymerization of the OEGMA resulting in a higher dispersity of molecular weights.

The composition and purity of the respective polymers were determined by hydrogen nuclear magnetic resonance (^1^H-NMR). Figure 6 shows the ^1^H-NMR spectrum of POEGMA, PDEGMA, and poly(OEGMA-*co*-DEGMA) varying its composition. The signal around 1 ppm is assigned to the –CH_3_ groups of the methacrylates and the signal at 1.8 ppm correspond to –CH_2_– of the main chain of OEGMA and DEGMA; the signal at 3.4 ppm corresponds –CH_3_ on the side chain of OEGMA and DEGMA (a); the signal at 3.5 ppm correspond to –CH_2_– (c and c′) and the signal at 4.2 ppm corresponds to –CH_2_– next to oxygen of the ester group in the side chain (b and b′).

The final mol fraction of each component was calculated by integration of the methylene groups in the side chains (signals between 3.5 and 3.8 ppm) without the adjacent to the ester group. This was taken such as reference with integration value of two (signal at 4.1 ppm) in the corresponding ^1^H-NMR spectrum. For only POEGMA the integration value was 15.54 (Appendix A) whereas for only PDEGMA was 6.00 (Appendix A). ^1^H-NMR spectra of the monomers of OEGMA and DEGMA are shown in Appendix A, respectively. Using these values is easy to calculate the final composition by solving the following two equation system.
*Int* = 15.54 × *χ*_OEGMA_ + 6.00 × *χ*_DEGMA_(4)
1 = *χ*_OEGMA_ + *χ*_DEGMA_(5)
where *Int* is the integration value measured in the ^1^H-NMR spectrum, *χ*_OEGMA_ and *χ*_DEGMA_ are the mol fraction of each component in the copolymer.

Therefore, the mol fraction of OEGMA can be calculated using the Equation (6).
*χ*_OEGMA_ = (*Int* − 6)/9.54(6)

The compositions of the synthesized copolymers were practically the same in the feed as in the final products (Figure 7) therefore an ideal copolymerization behavior is suggested. This means that the copolymerization parameters are close to one indicating the same kinetic rate for homo and cross-polymerization.

It is well known that the chemical structure and composition influence significantly the thermal properties of the copolymers. The glass transition temperature was measured by DSC. Figure 8 shows the thermograms for DEGMA and OEGMA polymers and its copolymers obtained at different composition. A linear increase on the *T_g_* values was clearly observed when the content of DEGMA was increased in the copolymer. The Figure 9 shows a linear relationship between DEGMA content and *T_g_* values, the linear regression had an *R*^2^ value of 0.9613.

The comparison between experimental results obtained by DSC and the predictions of the Gibbs-Di Marzio equation [33] is given in Table 1. The *T_g_* values obtained by DSC are very close to what is expected by the GD equation (Equation (7)) which means that random copolymers were obtained.
*T*_g_ = *χ*_OEGMA_ × *T*_g POEGMA_ + *χ*_DEGMA_ × *T*_g PDEGMA_(7)
where *χ*_OEGMA_ and *χ*_DEGMA_ are the corresponding mol fractions of OEGMA and DEGMA in the copolymer obtained.

Thermal stability was evaluated by thermogravimetric analysis under a nitrogen flow; decomposition temperature was similar in all cases, the temperature for 10% weight loss for PDEGMA was at 225 °C, for POEGMA at 226, for poly(OEGMA-*co*-DEGMA) 50:50 mole ratio was at 219, and for poly(OEGMA-*co*-DEGMA) 24:76 mole ratio was at 235 (Figure 10). These small differences may be because of polymers with more OEGMA had around 4% of water, even after desiccation due to more hydrophilic character.

### Thermo-Responsive Behavior

Many times, the acronym LCST is used referring to the temperature at which a phase separation occurs at one specific polymer concentration in a solvent. This is not correct due to the fact that the LCST is the minimum found in the bimodal curve obtained in a phase diagram for a binary mixture polymer-solvent [34], therefore some researchers usually use instead the more appropriate term “cloud point temperature” which is the temperature when the phase separation can be seen by simple observation of the mixture under heating (cloudiness). In this work *Tcp* is used referring to the temperature at which the phase transition was evaluated independent of the concentration and the used method.

Lutz reported that the “LCST” of aqueous solutions of copolymers based DEGMA and OEGMA (*M_n_* = 475 g mol^−1^) synthesized by ATRP, could be adjusted between 26 and 90 °C; furthermore the phase transition temperature was a linear function with respect to the OEGMA mol fraction in the obtained copolymer [29]. Wang and co-workers reported that copolymers of MEOMA and OEGMA (*M_n_* = 200 g mol^−1^) at different mole feed ratios, synthesized by free radical polymerization, also showed a tunable phase transition temperature which depended on the OEGMA mole ratio in the copolymer [35]. It is evident that the *Tcp* may be tuned using the correct monomers combination and by an adequate synthesis method. In this work we show that RAFT polymerization was an appropriate method to pursue this aim.

The *Tcp* was evaluated first by transmittance, a fast screening was done to each sample to observe the temperature where the dissolution turned from clear to opaque, then transmittance was measured some degrees below and this value was taken as 100%. Samples were heated and the *Tcp* was taken at the 50% drop in the transmittance.

In order to evaluate the effect of the copolymer composition, dissolution of polymers or copolymers at 2 mg mL^−1^ were prepared using distilled water. A fast change in transmittance was observed in response to very small change in the temperature around the *Tcp*; in most of the cases only one degree was enough to change the transmittance from 90 to 10%. The curves are shown in the Figure 11. Interesting, for the sample with 24% mol of DEGMA, the *Tcp* in water was 35.9 °C which is very close to the average human body temperature. Porch and coworkers synthesized a similar copolymer by ATRP and reported that the *Tcp* for a copolymer with 27% of DEGMA was at 38 °C [28]. Grishkewish and co-workers studied the same kind of polymers but grafted onto nanocrystalline cellulose, the phase transition temperature was also a function of the copolymer composition [36]. Nevertheless, there are other factors that affect this value for example the presence of salts and its concentration, Mangnusson and co-workers studied the effect of different salts in similar copolymers finding that the phase transition temperature depends of the salt type, however, phosphates were not studied [37]. Jones and co-workers observed that, for a copolymer OEGMA-methacrylic acid, the *Tcp* was higher when urea salt was added [31]. For that reasons, in this work, the *Tcp* was evaluated in distilled water and also in PBS. In all cases using PBS the *Tcp* was found a few degrees lower than using distilled water at the same polymer concentration. A linear relationship between *Tcp* and copolymer composition was found in both cases, in the Figure 12, we can observe that both lines are almost parallels. It is important to mention that the *Tcp* for a specific polymer or copolymer is not only function of polymer and salts concentrations, moreover it may depend on the molecular weight. Yamamoto and co-workers [19] and, Han and co-worker, for example [13] have observed that the higher the molecular weight, the lower the *Tcp*. To observe this effect in this work, the *Tcp* of PDEGMA with three different *M_n_* was evaluated by turbidimetry, the *Tcp* values for select samples with *M_n_* = 43,670, 37,500, and 12,600 were at 23.3, 24.7, and 25.5 °C, respectively, therefore the effect of the molecular weight was clearly observed. Other characteristics of the copolymer as architecture or branching play also an important role in the thermo-responsive behavior [38,39].

To observe the effect of the polymer concentration, the *Tcp* of poly(DEGMA-*co*-OEGMA) with 76:24 mole ratio, was evaluated in water and in PBS at concentrations between 1 and 20 mg mL^−1^. As mentioned before, the *Tcp* in all cases, at the same concentration, was lower when PBS was used as solvent (Figure 13a). However, in both cases the temperature of phase separation was slightly lower when the polymer concentration was increased. At concentrations from 1 to 20 mg mL^−1^ using PBS, the difference was just 1.2 degrees whereas in the same concentration range in water was of two degrees (Figure 13b). For PDEGMA in water its *Tcp* increased from 22.7 °C at 20 mg mL^−1^ to 25.2 at 2 mg mL^−1^ (Appendix A). For the copolymer with 86:14 mole ratio its *Tcp* increased three degrees at the same range of concentrations and for POEGMA the effect was a little bit higher due to its *Tcp* was 61.5 °C at 20 mg mL^−1^ and 66.5 at 2 mg mL^−1^. The effect of the polymer concentration has been also observed by Lutz and coworkers for DEGMA-OEGMA (*M_n_* = 475 g mol^−1^) based copolymers synthesized by ATRP, the difference in the same range was of 4 °C [29].

When the size was measured by DLS, a sudden size increase due to aggregates formation was observed (Figure 14). The onset points were taken as *Tcp* and these values were similar to the other ones measured by transmittance at the same concentration in the same solvent. For instance the sample of poly(DEGMA-*co*-OEGMA) with 76:24 mole ratio at concentration of 20 mg mL^−1^ in PBS increased its size at around 31.5 °C, the same value was observed by transmittance when the same sample at the same concentration in water was evaluated, the values were at 34.5 °C using both methods.

The same sample at 6 mg mL^−1^ was analyzed and compared. Size increase was observed at 35.0 °C whereas by transmittance the 50% drop of this value was at 35.3 °C. Others three samples of copolymers were selected and analyzed, the *Tcp* values measured by DLS were very close to those measured by transmittance (Table 2). With these results it is clear that the phase transition temperature can be measured using any of these methods obtaining similar values: By DLS taking the onset point of size increase and by transmittance taking the 50% drop of the value. The second one is the most used method due to its easy access and it may be carried out using a simple spectrophotometer; however, for more diluted samples perhaps is not be the best option due to the transmittance do not decrease low enough. 

## 4. Conclusions

RAFT polymerization of temperature responsive statistical copolymers based on OEGMA and DEGMA showed an ideal copolymerization behavior, this means that, the composition of the obtained copolymers, which was calculated by ^1^H-NMR, was the same as in the monomers feed. The copolymers were obtained with high yields (up to 95%) after only 4 h of reaction. The *T_g_* values obtained showed also an almost ideal behavior in accordance with the Gibbs-Di Marzio equation. As a result, the *Tcp* can be easily adjusted by changing the molar ratio of monomers in the feed. Using distilled water at 2 mg mL^−1^, the *Tcp* values were between 26 and 67 °C showing a linear increase with the mol fraction of OEGMA. The influence of the presence of phosphate salts (PBS) and polymer concentration on the *Tcp* was evaluated. As effect of PBS the *Tcp* values were lower than using distilled water. On the other hand, when the polymer concentration was higher the *Tcp* decreased between one and four degrees in the concentration range between 2 and 20 mg mL^−1^. The results obtained at 50% of transmittance were similar to the ones registered at the set point by DLS when size is increased due to agglomerates formation. Due to the above, polymers with adjustable *Tcp* can be obtained which may be used for different applications such as in drug delivery systems, sensors, micro-valves, to mention some.

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
