# Peer review of "Tunable Thermo-Responsive Copolymers from DEGMA and OEGMA Synthesized by RAFT Polymerization and the Effect of the Concentration and Saline Phosphate Buffer on Its Phase Transition"

_polymers, 2019, doi:10.3390/polym11101657_

Round 1
Reviewer 1 Report
The work seems to be of average to good quality from cursory look.
The manuscript is full of typos and too long statements. please revise carefully.
Revise
line 24-27
38, 46-52 ( really complex- please divide it into several simple statements)
line 55- spelling of temperature.
57- ih ??
63- compliation ??
107 - standars??
182-188 (one statement - really long)
196- firs ?/
I might have missed have some of them too - please revise it.
Author Response
Thank you very much for your prompt review.
The manuscript has been reviewed; the changes are highlighted in yellow color
Reviewer 2 Report
Comments: The authors synthesized a new trithiocarbonate-based RAFT agent which was used to polymerize thermos-responsive copolymers. This work is interesting and deserves publication. However, there are a few errors that should be addressed before publication. Hope the authors reflect the following comments and improve the work.
The writing and representation of paper can be improved. Specifically, there are many grammar errors and mistakes. In Page 2 line 55, “temperaturee” should be “temperature”. In Page 2 line 57, “ih” should be “in”. In Page 2 line 59, “this because” should be “this is because” The authors developed a new RAFT agent using APP. However, it was not clear why the authors don’t use the precursor of APP as a RAFT agent. It is known that the precursor of APP can also function as active RAFT agent. It is recommended that the author add more explanation of the motivation their synthesis of APP. In the Introduction, the authors should consider adding references when introducing the concept of ATRP. 1. Lei, Zhiwen, et al. "Thermoresponsive melamine sponges with switchable wettability by interface-initiated atom transfer radical polymerization for oil/water separation." ACS applied materials & interfaces 9.10 (2017): 8967-8974. 2. Luo, Jianhui, et al. "Electrostatic‐Driven Dynamic Jamming of 2D Nanoparticles at Interfaces for Controlled Molecular Diffusion." Angewandte Chemie 130.36 (2018): 11926-11931. 3. Yuan, Tianyu, et al. "Assembly and chiral memory effects of dynamic macroscopic supramolecular helices." Chemistry–A European Journal 24.62 (2018): 16553-16557.
Author Response
Thank you very much for you prompt review.
The manuscript has been reviewed; the changes are highlighted in yellow color.
With respect to the use of APP, this was due to the fact that of the precursor in not very stable even at low temperature, this is an orange liquid that hydrolyzes to form traces of APP affecting the polymerization; this is mentioned in the first paragraph of results and discussion.
The suggested papers are very interesting. However, we cited only some examples of works where PDEGMAS or POEGMAs were synthesized by ATRP.
Round 2
Reviewer 1 Report
Comments
Grammatical errors, line 26 Simplification of sentences in line 37 to 40 needs to be done. Introduce all abbreviations used in abstract within the abstract and add few words on the importance of this study in the abstract. What is Anal. calcd.??? Line 146. How the complete removal of solvent from synthesized polymer was ensured? Ratios used should be clearly mentioned in the beginning. What was the basis for selecting this ratio? M:CTA:I = 500:4:1 mole ratio. Figure 3 mentions ratio to be M:CTA:I = 500:45:1. Two ratios were tried? Not clear R2 value for best fit line should always be mentioned. In figure 4, which elution time represent which molecular weight is not clear, coding needs to be improved. Figure 8 thermograms, at peaks temperature, should have been mentioned. Coding should be uniform either in mole ratio or in mole %. Why different studies are done for different ratios of monomer? It is not clear. At some places Poly(OEGMA-co-DEGMA) ratio is mentioned as 25:75 and at some places 26:74, why? What are the overall importance and future prospects of this study?
Author Response
Grammatical errors, line 26
Answer: It was corrected and some sentences were interchanged.
Simplification of sentences in line 37 to 40 needs to be done.
Answer: Done.
Introduce all abbreviations used in abstract within the abstract and add few words on the importance of this study in the abstract.
Answer: Done.
What is Anal. calcd.??? Line 146.
Answer: It is the abbreviation used for Analytically Calculated; this refers to the elemental analysis.
How the complete removal of solvent from synthesized polymer was ensured?
Answer: By 1HNMR. The solvent signals were not observed.
Ratios used should be clearly mentioned in the beginning. What was the basis for selecting this ratio? M:CTA:I = 500:4:1 mole ratio. Figure 3 mentions ratio to be M:CTA:I = 500:45:1. Two ratios were tried? Not clear
Answer: The correct mole ratio is 500:4:1; this was selected based on previous reports where RAFT polymerization has been used. This because the higher the CTA:I ratio the better control but the longer reaction time.
R2 value for best fit line should always be mentioned.
Answer: R2 values were introduced
In figure 4, which elution time represent which molecular weight is not clear, coding needs to be improved.
Answer: The figure and caption were modified.
Figure 8 thermograms, at peaks temperature, should have been mentioned.
Answer: Tg’s were calculated by the software Universal Analysis from TA Instrument, the values are shown in the table 1. Tg’s were taken at the half of the slope change.
Coding should be uniform either in mole ratio or in mole %.
Answer: Done. Mole ratio is used.
Why different studies are done for different ratios of monomer? It is not clear. At some places Poly(OEGMA-co-DEGMA) ratio is mentioned as 25:75 and at some places 26:74, why?
Answer: Some changes were done. When the mole ratio refers to the feed one, it is specified in the text.
What are the overall importance and future prospects of this study?
Answer: It is mentioned now in the conclusions.